# Optimizing Antibiotic Therapy for *Stenotrophomonas maltophilia* Infections in Critically Ill Patients: A Pharmacokinetic/Pharmacodynamic Approach

**DOI:** 10.3390/antibiotics13060553

**Published:** 2024-06-13

**Authors:** Helena Barrasa, Miguel Angel Morán, Leire Fernández-Ciriza, Arantxa Isla, María Ángeles Solinís, Andrés Canut-Blasco, Alicia Rodríguez-Gascón

**Affiliations:** 1Intensive Care Unit, Araba University Hospital, Osakidetza Basque Health Service, 01009 Vitoria-Gasteiz, Spain; 2Bioaraba Health Research Institute, 01009 Vitoria-Gasteiz, Spain; miguelangel.moranrodriguez@osakidetza.eus (M.A.M.); arantxa.isla@ehu.eus (A.I.); marian.solinis@ehu.eus (M.Á.S.); andres.canutblasco@osakidetza.eus (A.C.-B.); 3Infectious Disease Division, Araba University Hospital, Osakidetza Basque Health Service, 01009 Vitoria-Gasteiz, Spain; 4Microbiology Laboratory, Biomedical Diagnostic Service, Hospital San Pedro, 26006 Logroño, Spain; lfciriza@riojasalud.es; 5Pharmacokinetic, Nanotechnology and Gene Therapy Group (PharmaNanoGene), Faculty of Pharmacy, Lascaray Research Centre, University of the Basque Country UPV/EHU, 01006 Vitoria-Gasteiz, Spain; 6Microbiology Service, Araba University Hospital, Osakidetza Basque Health Service, 01009 Vitoria-Gasteiz, Spain

**Keywords:** *Stenotrophomonas maltophilia*, PK/PD, cotrimoxazole, levofloxacin, minocycline, tigecycline, aztreonam/avibactam, cefiderocol

## Abstract

*Stenotrophomonas maltophilia* is an opportunistic, multidrug-resistant non-fermentative Gram-negative bacillus, posing a significant challenge in clinical treatment due to its numerous intrinsic and acquired resistance mechanisms. This study aimed to evaluate the adequacy of antibiotics used for the treatment of *S. maltophilia* infections in critically ill patients using a pharmacokinetic/pharmacodynamic (PK/PD) approach. The antibiotics studied included cotrimoxazole, levofloxacin, minocycline, tigecycline, cefiderocol, and the new combination aztreonam/avibactam, which is not yet approved. By Monte Carlo simulations, the probability of target attainment (PTA), the PK/PD breakpoints, and the cumulative fraction of response (CFR) were estimated. PK parameters and MIC distributions were sourced from the literature, the European Committee on Antimicrobial Susceptibility Testing (EUCAST), and the SENTRY Antimicrobial Surveillance Program collection. Cefiderocol 2 g q8h, minocycline 200 mg q12h, tigecycline 100 mg q12h, and aztreonam/avibactam 1500/500 mg q6h were the best options to treat empirically infections due to *S. maltophilia*. Cotrimoxazole provided a higher probability of treatment success for the U.S. isolates than for European isolates. For all antibiotics, discrepancies between the PK/PD breakpoints and the clinical breakpoints defined by EUCAST (or the ECOFF) and CLSI were detected.

## 1. Introduction

*Stenotrophomonas maltophilia* is an opportunistic, multidrug-resistant non-fermentative Gram-negative bacilli that causes a variety of clinical syndromes including hospital-acquired pneumonia (HAP) or ventilator-associated pneumonia (VAP) as well as bloodstream infections (BSIs), particularly in debilitated or immunocompromised patients, with high mortality rates [1]. *S. maltophilia* ranks among the most frequent pathogens isolated from hospitalized pneumonia patients in Europe, and it is the primary Gram-negative pathogen resistant to carbapenems isolated from BSIs in the USA [2]. *S. maltophilia* is commonly found in cystic fibrosis (CF) airways which can cause colonization and chronic infection in patients with CF [3].

According to the latest guidelines of The Infectious Diseases Society of America (IDSA) [4,5] for mild infections, the recommended treatment includes cotrimoxazole and minocycline either as monotherapy, or alternatively, tigecycline, levofloxacin, or cefiderocol as monotherapy. It is strongly advised to refrain from using ceftazidime due to its probable ineffectiveness, and improvement is not expected when combined with avibactam without aztreonam. For moderate to severe infections, at least three distinct approaches are suggested: (1) combination therapy comprising cotrimoxazole plus minocycline; (2) initial treatment with cotrimoxazole alone, with the subsequent addition of minocycline (preferred), tigecycline, levofloxacin, or cefiderocol if clinical improvement is delayed with cotrimoxazole alone; and (3) cefiderocol or ceftazidime/avibactam in conjunction with aztreonam, particularly when intolerance toward or ineffectiveness of other agents are anticipated.

*S. maltophilia* is intrinsically resistant to many classes of antibiotics, including aminoglycosides, quinolones, and beta-lactams, through the production of aminoglycoside-modifying enzymes (AMEs), Qnr-like resistance determinants (Sm*qnr* genes), and two inducible beta-lactamases, L1 and L2 [6,7]. L1 metallo-beta-lactamase hydrolyse penicillins, cephalosporins, and carbapenems, but not aztreonam, and are not inhibited by any beta-lactamase inhibitor (BLI). L2 is a class A cephalosporinase susceptible to inhibition by clavulanic acid and avibactam. The overexpression of efflux pumps also contributes to resistance to tetracyclines, quinolones, cotrimoxazole (trimethoprim/sulfamethoxazole), tigecycline, and polymyxins. *S. maltophilia* also acquire genes horizontally that confer resistance to cotrimoxazole (*sul* and *dfrA*) and beta-lactams (*bla*) [4,8].

Although the Clinical and Laboratory Standards Institute (CLSI) has established susceptibility interpretative criteria for ticarcillin/clavulanate, ceftazidime, cefiderocol, minocycline, levofloxacin, cotrimoxazole, and chloramphenicol [9], the European Committee on Antimicrobial Susceptibility Testing (EUCAST) has only established interpretative criteria for cotrimoxazole [10]. The scarcity of available antimicrobial agents with clinical breakpoints recognized, the poor correlation data between MICs and clinical outcomes, and the lack of updated pharmacokinetic/pharmacodynamic studies make it difficult to treat *S. maltophilia* infections, especially in immunocompromised and critically ill patients when the first-line antimicrobial, cotrimoxazole, is contraindicated.

The aim of our study was to evaluate the adequacy of commonly prescribed antibiotics for the treatment of *S. maltophilia* infections in critically ill patients, including cotrimoxazole (trimethoprim/sulfamethoxazole), levofloxacin, minocycline, tigecycline, and cefiderocol, using a pharmacokinetic/pharmacodynamic (PK/PD) approach. The new combination, aztreonam/avibactam, which is not yet approved, was also studied. Our goal was to provide clinicians with additional insights to enhance antibiotic therapy and to combat antimicrobial resistance.

## 2. Results

Figure 1 shows the probability of target attainment (PTA, or the probability that the specific value of the PK/PD index reaches the values associated with the efficacy at a certain MIC value) of all the antimicrobials included in the study and the MIC distribution of *S. maltophilia* against these antibiotics. For the aztreonam/avibactam combination, since the PTA for avibactam (%ƒT_>2.5 mg/L_ > 50%) is 100%, the joint PTA corresponds to the PTA for aztreonam (%*f*T*_>_*_MIC_ ≥ 60%). MIC distributions of *S. maltophilia* for cotrimoxazole (trimethoprim/sulfamethoxazole), levofloxacin, minocycline, tigecycline, cefiderocol, and aztreonam/avibactam are available from European isolates (reported by EUCAST [11]); for trimethoprim/sulfamethoxazole, minocycline, and levofloxacin, MIC values from U.S. isolates were also available [7].

The most relevant differences in the susceptibility profile depending on the geographical area (Europe vs. U.S.) were observed for trimethoprim/sulfamethoxazole; while the MIC_90_ of the U.S. isolates is 1 mg/L, European isolates present an MIC_90_ of 16 mg/L. By comparing the PTA values and the MIC distributions, the best options to treat infections due to *S. maltophilia* in Europe are cefiderocol, aztreonam/avibactam, a higher dose of tigecycline, and a higher dose of minocycline. In the U.S., trimethoprim/sulfamethoxazole is also a good option. The MIC distributions of U.S. isolates against tigecycline, cefiderocol, and aztreonam/avibactam were not available, and therefore, we could not compare the PTA with the susceptibility profile of U.S. isolates.

The PK/PD breakpoint, that is, the highest MIC value at which there is a high probability of target attainment (≥97.5%), can be estimated by representing the values of the PK/PD index (*f*AUC_24_/MIC or %*f*T*_>_*_MIC_) against the MIC values. The PK/PD breakpoint can be read directly from the intersection of the horizontal line at the PK/PD target and the lower limit of the 95% confidence interval (2.5% percentile). Figure 2 features the relationship between the values of the *f*AUC_24_/MIC and the MIC for cotrimoxazole (trimethoprim component). With the three dose levels analyzed, the PK/PD breakpoint was 0.5 mg/L.

Figure 3 shows the relationship between the PK/PD index of levofloxacin (*f*AUC_24_/MIC) and the MIC. According to these results, and considering the target (*f*AUC_24_/MIC ≥ 62), the PK/PD breakpoint is 0.5 mg/L for 500 mg q12h and 0.25 mg/L for 500 mg q24h and 750 mg q24h.

In Figure 4, the relationship between the PK/PD index of minocycline (*f*AUC_24_/MIC) and the MIC is featured. Considering the target (*f*AUC_24_/MIC ≥ 8.75), the PK/PD breakpoint is 0.5 mg/L and 1 mg/L for 100 mg q12h and 200 mg q12h, respectively.

Figure 5 shows the relationship between the PK/PD index of tigecycline (*f*AUC_24_/MIC), and the MIC is featured. Considering the target (*f*AUC_24_/MIC ≥ 0.9), the PK/PD breakpoint is 0.5 mg/L and 1 mg/L for 50 mg q12h and 100 mg q12h, respectively.

The relationship between the PK/PD index of cefiderocol (%*f*T*_>_*_MIC_) and the MIC is featured in Figure 6. Considering the target (%*f*T*_>_*_MIC_ ≥ 75%), the PK/PD breakpoint for the dosing regimen of 2 g q8h is 4 mg/L.

Figure 7 presents the relationship between the PK/PD index of aztreonam (%fT_>MIC_) and the MIC. Considering the target (%fT_>MIC_ ≥ 60% for aztreonam), the PK/PD breakpoint for the dosing regimen of 1500 mg q6h is 2 mg/L.

Table 1 summarizes the PK/PD breakpoints calculated and the breakpoints published by CLSI [9] and EUCAST [10,11].

By Monte Carlo simulations, we also estimated the cumulative fraction of response (CFR), defined as the expected population PTA for a specific drug dose and a specific population of microorganisms. The CFR is considered the expected probability of success of a dosing regimen against bacteria when a specific value of MIC is not available, and thus, the population distribution of MICs is used. Table 2 shows the CFR values for all antimicrobials included in the study calculated from the corresponding MIC distribution of *S. maltophilia* isolates.

Considering the EUCAST MIC distribution, cotrimoxazole did not provide CFR values >90% at any of the dose levels analyzed. However, for the U.S. isolates, CFR ≥ 90% was obtained with all of the doses. For levofloxacin, regardless of the dose level and the precedence of the isolates, CFR was always <60%. Regarding minocycline, only the high dose provided CFR > 90%, regardless of the precedence of the isolates. Tigecycline only provided CFR > 90% at the high dose (100 mg q12h) for the EUCAST isolates. With cefiderocol and aztreonam/avibactam, CFR was 99% and 95%, respectively.

## 3. Discussion

Currently, the management of *S. maltophilia* infections remains a challenge due, among other factors, to (1) the significant number of intrinsic and acquired resistance mechanisms it presents, (2) the occasional difficulty in distinguishing colonization from a true infection, (3) the common polymicrobial presentation (specially in immunocompromised patients), (4) the absence of a standard of care to assess the effectiveness of therapeutic alternatives, and (5) the challenge in determining the sensitivity profile [4]. In this regard, adequate therapy is essential for survival. In a recent meta-analysis, patients with bacteremia treated with inappropriate antimicrobial therapy had higher mortality [12]. Apart from the selection of the most appropriate antibiotic, the success of therapy depends on optimizing the dosing regimen. The integration of pharmacokinetic/pharmacodynamic (PKPD) analysis and Monte Carlo simulation (MCS) has made them important tools for optimizing antimicrobial therapy. They enable the selection of the optimal antibiotic and dosing regimen taking into account the microorganism and the patient, increasing the chance of therapy success and reducing side effects and the emergence of resistant strains [13,14,15].

In this work, we have identified minocycline 200 mg q12h, tigecycline 100 mg q12h, cefiderocol 2 g q8h, and aztreonam/avibactam 1500/500 mg q6h as the best options to treat infections due to *S. maltophilia* considering the susceptibility of European isolates. For U.S. strains, cotrimoxazole at any dose level and minocycline 200 mg q12h are also good options; the lack of availability of U.S. isolate susceptibility profiles to tigecycline, cefiderocol, and aztreonam/avibactam prevented us from demonstrating the usefulness of these antibiotics. To the knowledge of the authors, this is a more complete study that evaluates, through PK/PD and Monte Carlo simulations, if the antimicrobials used for the treatment of infections due to *S. maltophilia* are adequate considering the susceptibility profile of this microorganism.

The highest MIC value at which a high probability of target attainment (PTA ≥ 90%) is obtained can be used to estimate PK/PD breakpoints; however, a much more restrictive PK/PD breakpoint can be obtained from a graphical representation of the PK/PD index as a function of the MIC (EUCAST approach) [16]. PK/PD breakpoints may be especially useful when no clinical breakpoint nor epidemiological cut-off values (ECOFFs) are defined, as is the case of *S. maltophilia*. Contrary to clinical breakpoints, different PK/PD breakpoints can be obtained with different dosages of the same drug, since they are dose regimen-dependent and species-independent [17]. Regarding cotrimoxazole, we detected discrepancies between the PK/PD breakpoint (0.5 mg/L) and the clinical breakpoint defined by EUCAST (susceptible, increased exposure up to 4 mg/L) and CLSI (susceptible up to 2 mg/L) (Table 1). According to the PK/PD breakpoint, cotrimoxazole should not be recommended to treat an infection due to isolates with MICs higher than 0.5 mg/L since drug exposure is insufficient. These results agree with Lasko et al. [18], who demonstrated that cotrimoxazole in monotherapy even at a high dose displays limited activity against cotrimoxazole-susceptible *S. maltophilia* strains. In fact, several authors have pointed out the necessity of redefining the breakpoints of cotrimoxazole for *S. maltophilia* [8,19,20]. Apart from cotrimoxazole, we detected discrepancies between the breakpoints for the rest of the antibiotics. Only with tigecycline and cefiderocol, the PK/PD breakpoints were higher than the ECOFF and the clinical breakpoint by CLSI, respectively; that is, antibiotic exposure is sufficient to treat infections due to isolates with MIC values considered resistant. The PK/PD breakpoints of tigecycline obtained in our study are in agreement with a systematic review, which pointed out that for an MIC < 0.5 mg/L, standard dosing of 50 mg q12h with a loading dose of 100 mg can be used, but if the MIC is ≥0.5–1 mg/L, a loading dose of 200 mg and 100 mg q12h is recommended [21]. Discrepancies in the PK/PD and clinical breakpoints have been already detected for other antibiotics [7,22]. In a previous study, the breakpoints of minocycline against *Acinetobacter baumanii* have been questioned since a modern PK/PD evaluation was not carried out for tetracyclines [23].

For empirical therapy, the susceptibility profile of the microorganism responsible for the infection in a certain geographical area is crucial. In this regard, a great difference in the susceptibility of *S. maltophilia* to cotrimoxazole between Europe and the U.S. exists, and therefore, the dose level needed may be different. According to our study, while in the U.S., a high level of treatment success is achieved with any of the three dose regimens studied (CFR ≥ 90%), in Europe, even with the highest dose evaluated, the probability of treatment success is much lower (CFR < 75%). In spite of our results, and despite the fact that a recent meta-analysis has shown that the resistance rate of *S. maltophilia* has increased in recent years [24], cotrimoxazole remains the first choice for treatment, with the dosing recommendation ranging from 10 to 15 mg/kg/day of trimethoprim q8h and a maximum daily dose of trimethoprim of 960 mg [1,8].

According to IDSA guidelines [4], minocycline and tigecycline are considered for the treatment of *S. maltophilia* infections. Although minocycline is preferred over tigecycline due to more favorable in vitro data, defined CLSI breakpoints, the availability of an oral formulation, and improved tolerability, our PK/PD study has revealed that both antibiotics at the highest dose (200 mg q12h of minocycline and 100 mg q12h of tigecycline) provide a high probability of treatment success (CFR > 90%) regardless of the geographical area, Europe or the U.S. It is noteworthy that tetracycline derivatives exhibit rapid tissue distribution after administration, leading to low concentrations in urine and serum [25]. Consequently, these derivatives are not advised as treatments of urinary tract infections (UTIs) due to *S. maltophilia* and are only recommended in combination therapy for the management of *S. maltophilia* bloodstream infections [4].

The role of quinolones in the treatment of *S. maltophilia* infections is controversial. While they may serve as an alternative in mild infections, their use in moderate to severe infections is only advised in combination therapy, with monotherapy de-escalation not recommended. This recommendation is based on suboptimal results in in vitro studies [26], low susceptibility in surveillance studies [27], the well-known resistance mechanisms (Sm*qnr* genes), the increasing development of resistance during treatment [24], adverse effects [28], and the lack of high-quality clinical studies supporting their use [29]. Our study has confirmed that levofloxacin is not a good alternative for the empirical treatment of infections due to *S. maltophilia*, neither in Europe nor in the U.S., at least in monotherapy. Only if the MIC is known and up to 0.5 mg/L, levofloxacin at a dose of 500 mg q12h could be an option; if the MIC is ≤0.25 mg/L, 500 mg or 750 mg q24 could also be useful. A previous PK/PD study also suggested than levofloxacin monotherapy may not achieve appropriate PK/PD target attainment for *S. maltophilia* infections [30]. Additionally, in a neutropenic murine tight infection model, levofloxacin 750 mg q24h provided probabilities of target attainment for a 1 − log_10_ CFU (colony-forming unit) reduction of 95.8%, 72.2%, and 26% at MICs of 0.5, 1, and 2 mg/L, respectively [31]. As with cotrimoxazole, it has been suggested that the CLSI breakpoint of levofloxacin (2 mg/L) should be redefined. In spite of the less favorable results obtained with levofloxacin, its usefulness for the treatment of *S. maltophilia* infections is controversial due to the lack of robust evidence supporting the superiority of one therapy over others. In this regard, in a recent meta-analysis, significant differences in the mortality of patients with *S. maltophilia* bacteremia among trimethoprim/sulfamethoxazole, fluoroquinolones, and minocycline were not found [12].

In a previous study, cefiderocol and aztreonam/avibactam have been identified as two of the most promising options of treatment for extensively drug-resistant (XDR) *S. maltophilia* infections with resistance to the preferred first-line antimicrobials [21]. Our results are in line with this statement. The susceptibility profile of *S. maltophilia* to cefiderocol, a new, recently approved catechol-conjugated cephalosporin, is very favorable, with global susceptibility higher than 95% [8,32]. Our PK/PD study has confirmed its usefulness for the treatment of infections due to *S. maltophilia* isolates with MICs up to 4 mg/L, and also for empirical treatment, with a probability of success around 100%. These results are in agreement with a recent meta-analysis, which has shown that cefiderocol and minocycline can be considered the preferred treatment alternatives for *S. maltophilia* infections due to low resistance rates, although regional differences in resistance rates to other antibiotics should be considered [33]. Although cefiderocol has shown clinical and microbiological success in some studies [34,35], further research to confirm its efficacy and safety in the real world, that is, in everyday use, is needed.

The combination of aztreonam/avibactam, currently in phase III clinical trials but not yet approved, has shown a global susceptibility rate against *S. maltophilia* of around 98%, including isolates not susceptible to cotrimoxazole [36]. This new beta-lactam/beta-lactam inhibitor combination is an alternative to the use of aztreonam plus ceftazidime/avibactam, which has demonstrated treatment success in complex pancreatic focus [37], bacteremia [38], and pneumonia [39] caused by *S. maltophilia*. While ceftazidime is a substrate of L1 beta-lactamase, it cannot hydrolyze aztreonam, and although L2 hydrolyzes aztreonam, it is inhibited by avibactam. In this way, aztreonam successfully reaches the penicillin-binding proteins (PBPs) of *S. maltophilia*, likely PBP3 [8]. The efficacy and safety of aztreonam/avibactam is being studied, with positive results reported in the interim analysis of two phase III clinical trials [40,41,42]. PK/PD analysis, at the approved doses for the aztreonam/avibactam combination (1.5/0.5 g q6h), has shown an optimal coverage for infections due to isolates with MICs up to 2 mg/L, and when used empirically, a high probability of treatment success is expected (CFR of 95%). These results confirm that aztreonam/avibactam may represent a valuable alternative for the treatment of *S. maltophilia* infections, addressing a major unmet medical need.

Our study presents several limitations. First, we have evaluated the usefulness of the antimicrobials by PK/PD considering its administration as monotherapy, although clinical guides recommend combination therapy in some situations, mainly to treat moderate to severe infections. Nevertheless, in spite of the fact that in vitro PK/PD studies have revealed that monotherapy is hardly bactericidal against *S. maltophilia* even with susceptible strains, combination therapy does not seem to significantly increase the antibacterial activity [20]. In a systematic review about novel therapies for the treatment of *S. maltophilia* infections, Gibb et al. advise against the routine use of antimicrobial combinations for pneumonia, catheter-associated bacteremia, urinary tract infection, or bacteremia [21]. In the case of abdominal perforation with an undrained abscess, two active agents may be used, and in the case of endocarditis, although there is not enough evidence, combination therapy has been used historically; therefore, dose recommendations based in our results may be not applicable in abdominal perforation and endocarditis. A recent systematic review and meta-analysis to compare the effects of monotherapy and combination therapy on the mortality of patients with *S. maltophilia* infections [43] concluded that combination therapy may have a role in the treatment of severe or complex cases of *S. maltophilia*; however, monotherapy resulted in having more favorable outcomes in terms of mortality in patients with hospital-acquired *S. maltophilia* pneumonia. Due to the low number of studies involved in the meta-analysis (only four), the authors suggest a longitudinal study to further explore this association. The second limitation is that we have the PK/PD analysis to critically ill patients, and the results may not be representative of other patient populations. In spite of these limitations, PK/PD analysis and Monte Carlo simulations are very useful tools to estimate susceptibility breakpoints, mainly when clinical breakpoints are not defined, and to predict the success of antimicrobial agents when used empirically considering the susceptibility profile in a certain geographical area.

## 4. Materials and Methods

This simulation study was based on literature data only, and therefore, no ethics approval was required.

### 4.1. Pharmacokinetic Parameters, PK/PD Targets, and Susceptibility Data

The PK parameters of trimethoprim, levofloxacin, minocycline, tigecycline, cefiderocol, aztreonam, and avibactam in critically ill patients, as well as the PK/PD targets (PD endpoints), were obtained from the literature. The PK parameters and the PK/PD targets are presented in Table 3.

The PK/PD target of cotrimoxazole (*f*AUC_24_/MIC ≥ 67.4) refers to the trimethoprim component; however, it was estimated in an in vitro model with trimethoprim and sulfamethoxazole in combination and not separately, and therefore, the activity of the two components is considered [18].

The MIC distributions of trimethoprim/sulfamethoxazole (13 distributions, 2511 observations), levofloxacin (10 distributions, 1979 observations), minocycline (8 distributions, 432 observations), tigecycline (5 distributions, 289 observations), cefiderocol (5 distributions, 338 observations), and aztreonam/avibactam (4 distributions and 805 observations) against *S. maltophilia* were obtained from EUCAST [11]. Additionally, the MIC profile of cotrimoxazole, levofloxacin, and minocycline was also obtained from the study by Pfaller et al. [7], who included 1522 *S. maltophilia* isolates from the SENTRY Antimicrobial Surveillance Program collection, representing 35 U.S. medical centers from 2014 to 2021.

### 4.2. Pharmacokinetic/Pharmacodynamic Analysis and Monte Carlo Simulation

A 10,000-subject Monte Carlo simulation was conducted for each antibiotic using Oracle^®^ Crystal Ball Fusion Edition v.11.1.2.3.500 (Oracle USA Inc., Redwood City, CA, USA). The PK/PD index and the magnitude or value of the PK/PD index associated with the success of therapy for every antibiotic are listed in Table 3, as well the dosing regimens studied. For trimethoprim, levofloxacin, minocycline, and tigecycline, *f*AUC_24_/MIC is the PK/PD index that best predicts efficacy, and it was estimated by non-compartmental analysis using the following equation [22]:*f*AUC_24_/MIC = D × Fu/CL where D is the daily dose, Fu is the unbound fraction, and CL is the total clearance.

For cefiderocol and aztreonam, the proportion of time that the unbound serum concentration remains above the MIC in a steady state (%*f*T_>MIC_) is the PK/PD index related to efficacy, and it was calculated considering a one-compartmental pharmacokinetic model with the following equations [22]:$$t_{1}=\frac{MIC-fC_{min,ss}}{fC_{max,ss}-fC_{min,ss}}\cdot t_{inf}$$
$$t_{2}=\left(\frac{fC_{max,ss}}{MIC}\right)\cdot \frac{V_{ss}}{{CL}_{t}}$$
$${f\%T}_{>MIC}=\left[\left(t_{2}+t_{inf}\right)-t_{1}\right]\cdot \frac{100}{\tau }$$
where *f*C_min,ss_ and *f*C_max,ss_ are the minimum and the maximum serum concentration of the unbound drug (mg/L) in the steady state, respectively, t_inf_ is the infusion length, and t_1_ and t_2_ are the times at which the antibiotic concentration reaches the MIC during the infusion phase and in the elimination phase, respectively.

In the case of avibactam, the proportion of the dosing interval for which the free drug concentration is >2.5 mg/mL was calculated.

For simulations, a log-normal distribution was assumed for pharmacokinetic parameters, according to statistical criteria. The unbound fraction was included as a fixed value if interindividual variability was not available [55].

For each MIC value, the simulation resulted in a probability distribution, accounting for the variability in the pharmacokinetic parameters. The mean value and the 95% confidence interval (represented as percentiles) of either %*f*T_>MIC_ or *f*AUC_24_/MIC were then extracted. The highest MIC values at which the PK/PD index reaches the value related to treatment success, that is, the PK/PD breakpoints, were estimated from the lower limit of the 95% CI (2.5% percentile), as EUCAST has established [16].

We estimated the PTA or probability that a PK/PD index reaches the target value, that is, the value associated with efficacy [56]. PTA ≥ 90% was associated with a high probability of treatment success, but if 80% ≤ PTA < 90%, only a moderate probability of treatment success was considered [22].

The cumulative fraction of response, or CFR, that is, the probability of success of an empirical treatment, was estimated from the PTA at every MIC level and the MIC distribution of the bacterial population. For CFR calculation, the following equation was used [56]:$$CFR\left(\%\right)=\sum_{i=1}^{n}{PTA}_{i}\cdot F_{i}$$
where PTA_i_ is the PTA at each MIC value (i), and F_i_ is the percentage of the microorganisms in the population with this MIC value. As with PTA, for CFR ≥ 90%, we considered a high probability of treatment success, but it was only moderate if 80% ≤ CFR < 90% [22].

Joint PTA, defined as the simultaneous attainment of PTA, was calculated for aztreonam/avibactam [57]. We first determined if the PTA of avibactam was achieved, and if the threshold was met, we then estimated the joint PTA as the PTA calculated for aztreonam.

## 5. Conclusions

From a PK/PD perspective, the best options to treat empirically *S. maltophilia* infections are cefiderocol, aztreonam/avibactam, and minocycline and tigecycline at the highest doses. While for European isolates, cotrimoxazole provided a low probability of treatment success, this antimicrobial agent may be useful considering the susceptibility of U.S. isolates. For all antibiotics, discrepancies between the PK/PD breakpoints and the clinical breakpoints defined by EUCAST (or the ECOFF) and CLSI were detected. With cotrimoxazole, levofloxacin, minocycline, and aztreonam/avibactam, isolates were considered susceptible based on CLSI and EUCAST breakpoints, and the PK/PD analysis predicted insufficient exposure. Only with tigecycline and cefiderocol were the PK/PD breakpoints higher than the ECOFF and the clinical breakpoint by CLSI, respectively; that is, antibiotic exposure is sufficient to treat infections due to isolates considered resistant.

## Figures and Tables

**Figure 1 antibiotics-13-00553-f001:**
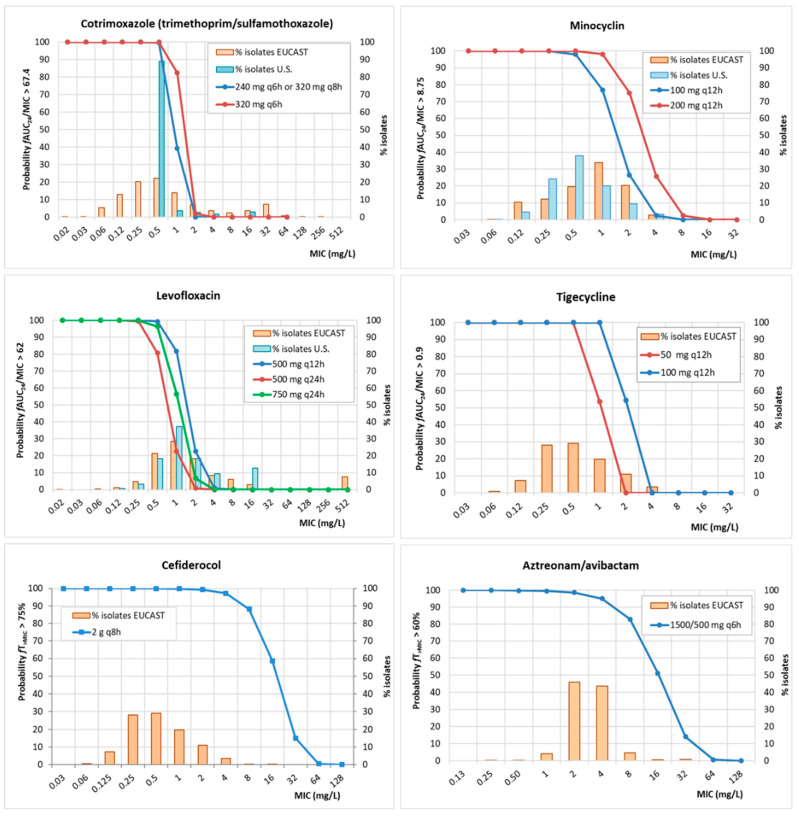
Probability of target attainment (PTA) and MIC distribution of *S. maltophilia* against cotrimoxazole, levofloxacin, minocycline, tigecycline, cefiderocol, and aztreonam/avibactam. For aztreonam/avibactam, PTA corresponds to the joint PTA (target of aztreonam and target for avibactam achieved simultaneously). For cotrimoxazole, the PTA refers to trimethoprim.

**Figure 2 antibiotics-13-00553-f002:**
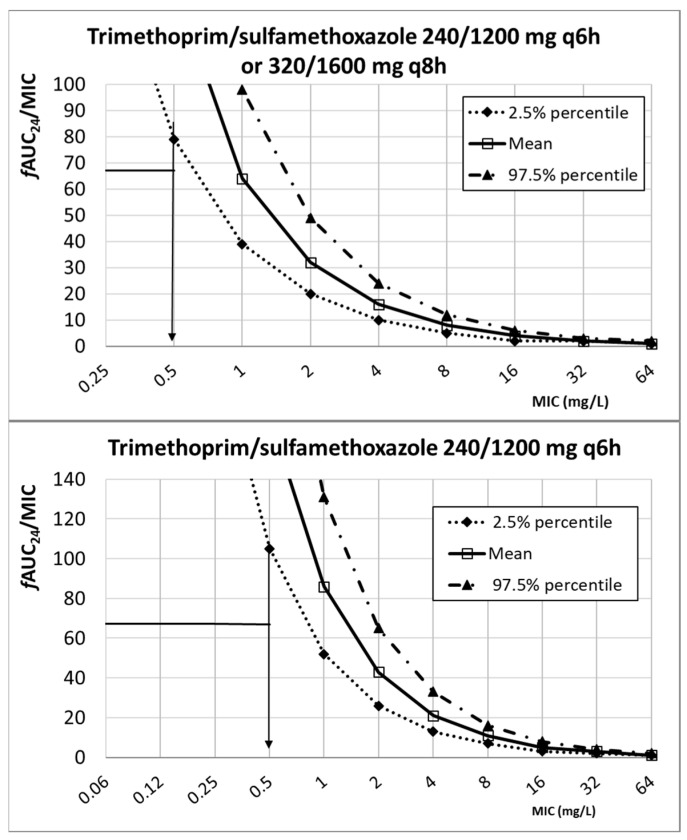
Relationship between *f*AUC_24_/MIC and MIC for different dosing regimens of cotrimoxazole (trimethoprim component). PK/PD target (horizontal line): 67.4.

**Figure 3 antibiotics-13-00553-f003:**
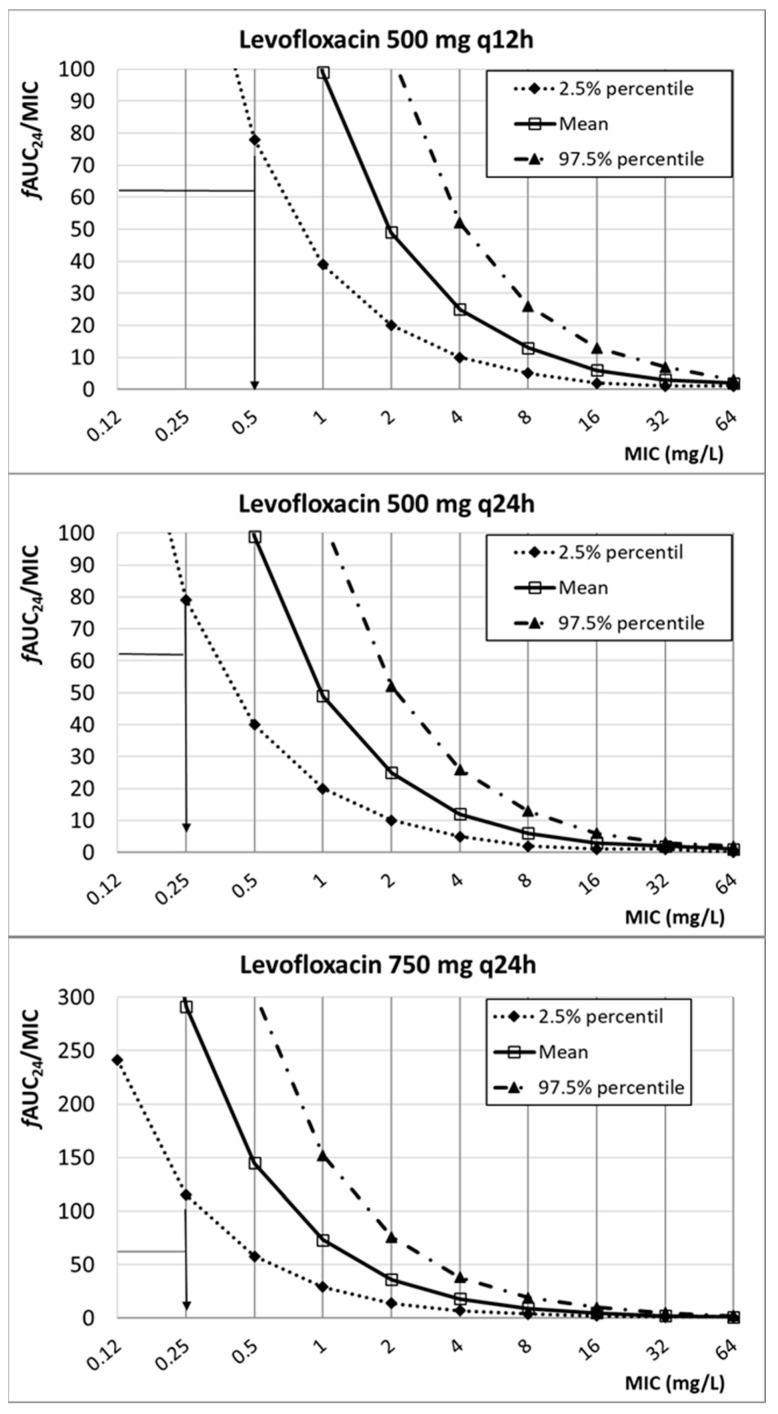
Relationship between *f*AUC_24_/MIC and MIC for different dosing regimens of levofloxacin. PK/PD target (horizontal line): 62.

**Figure 4 antibiotics-13-00553-f004:**
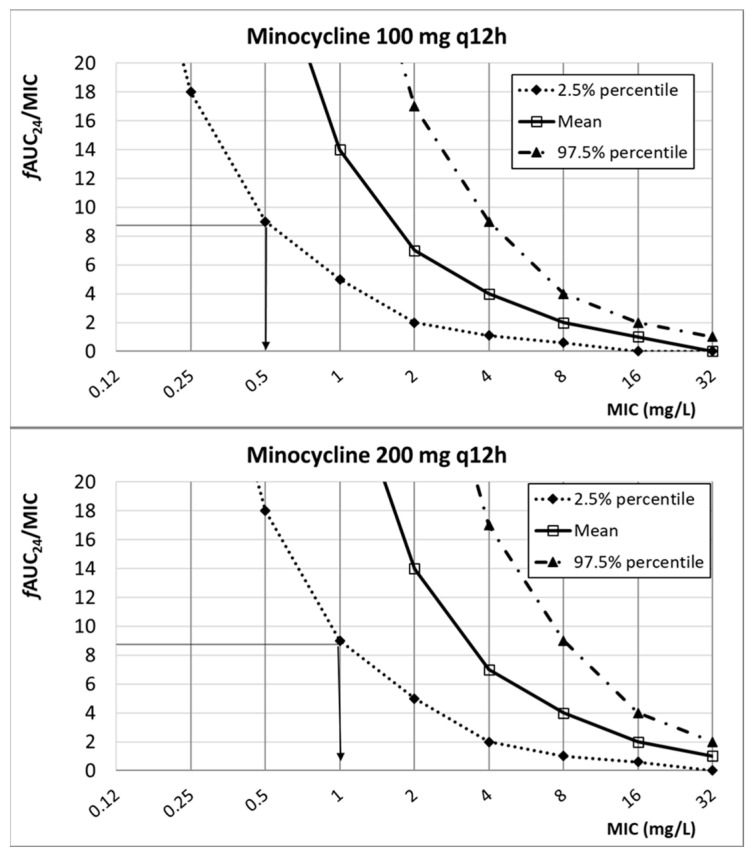
Relationship between *f*AUC_24_/MIC and MIC for different dosing regimens of minocycline. PK/PD target (horizontal line): 8.75.

**Figure 5 antibiotics-13-00553-f005:**
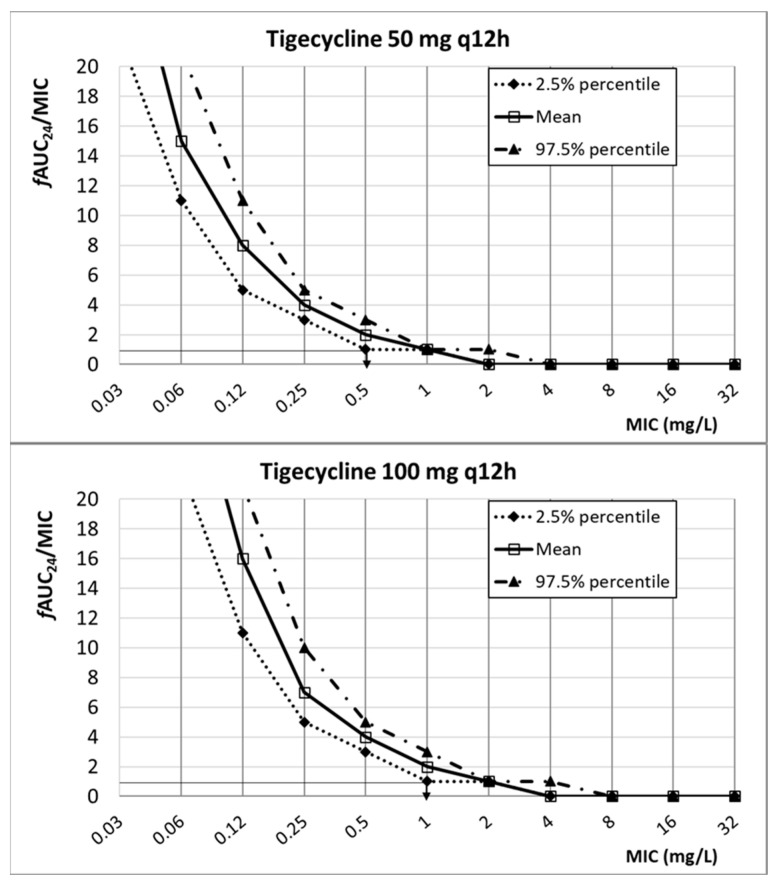
Relationship between *f*AUC_24_/MIC and MIC for different dosing regimens of tigecycline. PK/PD target (horizontal line).

**Figure 6 antibiotics-13-00553-f006:**
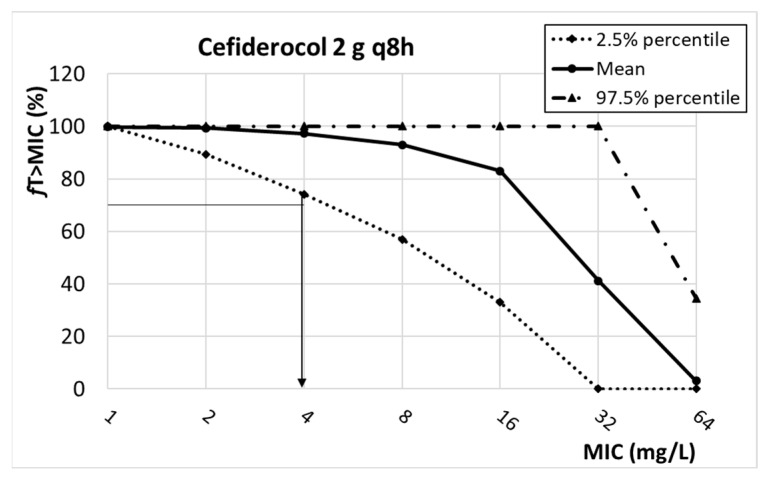
Relationship between *f*AUC_24_/MIC and MIC for cefiderocol. PK/PD target (horizontal line): 75%.

**Figure 7 antibiotics-13-00553-f007:**
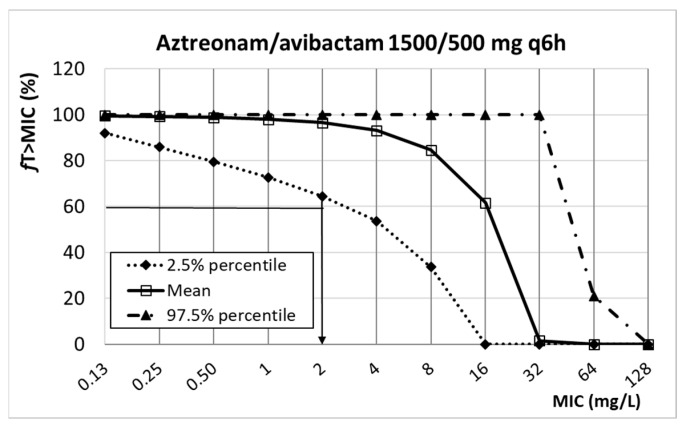
Relationship between %*f*T*_>_*_MIC_ and MIC for aztreonam/avibactam. PK/PD target (horizontal line): 60%.

**Table 1 antibiotics-13-00553-t001:** Comparison of the pharmacokinetic/pharmacodynamic (PK/PD), European Committee on Antimicrobial Susceptibility Testing (EUCAST), and Clinical and Laboratory Standards Institute (CLSI) breakpoints (mg/L) for *S. maltophilia*.

Antibiotic	PK/PD Breakpoint	Clinical Breakpoint		ECOFFs ^a^
		CLSI	EUCAST	
Cotrimoxazole (trimethoprim/sulfamethoxazole)		≤2 ^b^	≤4 ^b,c^	2
240/1200 mg q6h	0.5			
320/1600 mg q8h	0.5			
320/1600 mg q6h	0.5			
Levofloxacin		≤2	n.d.	4
500 mg q24h	0.25			
500 mg q12h	0.5			
750 mg q24h	0.25			
Minocycline		≤4	n.d.	1
100 mg q12h	0.5			
200 mg q12h	1			
Tigecycline		n.d.	n.d.	4
50 mg q12h	0.5			
100 mg q12h	1			
Cefiderocol		≤1	n.d.	0.125
2 g q8h	4			
Aztreonam/avibactam		n.d.	n.d.	8 ^d^
1500/500 mg q6h	2			

^a^: Epidemiological cut-off values according to EUCAST; ^b^: trimethoprim component; ^c^: susceptible, increased exposure: ^d^: tentative ECOFF (TECOFF). n.d.: not defined.

**Table 2 antibiotics-13-00553-t002:** CFR values of all antimicrobials included in the study for EUCAST and U.S. isolates.

Antibiotic	CFR (%)	
	EUCAST Isolates	U.S.Isolates
Cotrimoxazole (trimethoprim/sulfamethoxazole)		
240/1200 mg q6h	66	90
320/1600 mg q8h	66	90
320/1600 mg q6h	73	91
Levofloxacin		
500 mg q24h	30	28
500 mg q12h	55	57
750 mg q24h	44	44
Minocycline		
100 mg q12h	74	84
200 mg q12h	92	94
Tigecycline		
50 mg q12h	76	*
100 mg q12h	91	*
Cefiderocol		
2 g q8h	99	*
Aztreonam/avibactam		
1500/500 mg q6h	95	*

*: no MIC distribution was available, and therefore, CFR was not estimated.

**Table 3 antibiotics-13-00553-t003:** Pharmacokinetic parameters and PK/PD targets of the antimicrobial agents used for Monte Carlo simulations. Data are expressed as mean ± standard deviation.

Antibiotic	Dosing Regimen	CL (L/h)	Fu	V (L)	PK/PD Target	References
Cotrimoxazole (trimethoprim/sulfamethoxazole) ^a^	240/1200 mg q6h 320/1600 mg q6h 320/1600 mg q8h	1.88 ± 0.44 ^b^	0.5 ± 0.017		*f*AUC_24_/MIC ≥ 67.4	[18,44,45]
Levofloxacin	500 mg q12h 500 mgq24h 750 mgq24h	8.66 ± 3.85	0.71		*f*AUC_24_/MIC ≥ 62	[30,46,47]
Minocycline	100 mg q12h 200 mgq12h	4.70 ± 2.14	0.28		*f*AUC_24_/MIC ≥ 8.75	[30,48]
Tigecycline	50 mg q12h 100 mg q12h	22.10 ± 3.82	0.2		*f*AUC_24_/MIC ≥ 0.9	[30,49]
Cefiderocol	2 g q8h, 3 h infusion	4.04 ± 1.52	0.44 ± 0.04	V1: 7.78 ± 4.43 V2: 5.77 ± 1.94 V3: 0.798	%*f*T_>MIC_ > 75%	[50,51]
Aztreonam/ avibactam	1500/500 mg q6h 3 h infusion	9.60 ± 5.00 11.09 ± 6.78	0.72 0.92	27.20 ± 20.80 50.81 ± 14.32	%*f*T_>MIC_ > 60% %ƒT_>2.5 mg/L_ > 50%	[52,53,54]

^a^: pharmacokinetic parameters and PK/PD index refer to trimethoprim. ^b^: mL/min/Kg (70 Kg body weight was considered). CL: total body clearance; Fu: unbound fraction; *f*AUC_24_: area under the unbound concentration–time curve over a period of 24 h; %*f*T_>MIC_: percentage of time of the dosing interval in which the unbound serum antibiotic concentration remains above the minimum inhibitory concentration (MIC); V: distribution volume.

## Data Availability

All data are contained within the paper.

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
