# Peer review of "Optimizing Antibiotic Therapy for Stenotrophomonas maltophilia Infections in Critically Ill Patients: A Pharmacokinetic/Pharmacodynamic Approach"

_antibiotics, 2024, doi:10.3390/antibiotics13060553_

Round 1

Reviewer 1 Report

Comments and Suggestions for Authors
    • Dear Authors,
    • Your manuscript is clear and well-structured.
    • Thank you for this article regarding antibiotics strategy against S. maltophilia,  as a multidrug-resistant bacterium. Still, she is related to high mortality and unclearness about which therapy is better - monotherapy or a combination of antibiotics drugs. So, PKPD modeling is very useful for future decision processes. Besides, you should give more available literature data related to novel treatment strategies for S.maltophilia and compare results with the authors. Gibb et al. have shown a spectrum of novel antimicrobial treatments.  Also, it is interesting to compare combination versus monotherapy – there is a systematic review and meta-analysis (Prawang et al., 2022). You could improve the Discussion if you include those results.
    •  Data are shown with tables and they are easy to understand. Also, results are reproducible and interpreted appropriately and consistently
    • Are the manuscript’s results reproducible based on the details given in the methods section? The conclusion is based on strong arguments (MIC, PK/PD index).
    • The literature could be improved by the addition of relevant references.
  •  

Author Response

Point-by-point responses to reviewer 1 (changes in the manuscript are highlighted in red)

...you should give more available literature data related to novel treatment strategies for S.maltophilia and compare results with the authors. Gibb et al. have shown a spectrum of novel antimicrobial treatments.  Also, it is interesting to compare combination versus monotherapy – there is a systematic review and meta-analysis (Prawang et al., 2022). You could improve the Discussion if you include those results.

We thank the author for this comment. We have implemented the discussion accordingly, and added the recommended references in the reference list. The new paragraphs are the following:

Page 11, first paragraph: In this regard, adequate therapy is essential for survival. In a recent meta-analysis, patients with bacteremia treated with inappropriate antimicrobial therapy had higher mortality [12].

Page 12, first paragraph: The PK/PD breakpoints of tigecycline obtained in our study are in agreement to a systematic review, which pointed out that for MIC < 0.5 mg/L, standard dosing of 50 mg q12 h with loading dose of 100 mg can be used, but if MIC is ≥ 0.5-1 mg/L, loading dose of 200 mg and 100 mg q12 h is recommended [21].

Page 13, first paragraph: In spite of the less favorable results obtained with levofloxacin, its usefulness for the treatment of S. maltophilia infections is controversial due to lack of robust evidence supporting the superiority of one therapy over others. In this regard, in a recent meta-analysis, significant differences in mortality of patients with S. maltophilia bacteremia among trimethoprim-sulfamethoxazole, fluoroquinolones and minocycline were not found [12].

Page 13, second paragraph: In a previous study, aztreonam-avibactam and cefiderocol have been identified as two of the most promising options of treatment for extensively drug-resistant (XDR) S. maltophilia infections with resistance to the preferred first-line antimicrobials [21]. Our results are in line with this statement.

Page 14, first paragraph: In a systematic review about novel therapies for the treatment of S. maltophilia infections, Gibb et al advice against the routine use of antimicrobial combinations for pneumonia, catheter-associated bacteremia, urinary tract infection or bacteremia [21]. In case of abdominal perforation with undrained abscess, two active agents may be used, and in case of endocarditis, although there is not enough evidence, combination therapy has been used historically; therefore, dose recommendations based in our results may be not applicable in abdominal perforation and endocarditis. A recent systematic review and meta-analysis to compare the effects of monotherapy and combination therapy on mortality of patients with S. maltophilia infections [43] concluded that combination therapy may have a role in the treatment of severe or complex cases of S. maltophilia; however, monotherapy resulted to have more favorable outcomes in terms of mortality in patients with hospital acquired S. maltophilia pneumonia. Due to low number of studies involved in the meta-analysis (only 4), the authors suggest a longitudinal study to further explore this association.

Page 11, first paragraph: In this regard, adequate therapy is essential for survival. In a recent meta-analysis, patients with bacteremia treated with inappropriate antimicrobial therapy had higher mortality [12].

  • Data are shown with tables and they are easy to understand. Also, results are reproducible and interpreted appropriately and consistently.

We thanks the reviewer for this comment.

  • Are the manuscript’s results reproducible based on the details given in the methods section? The conclusion is based on strong arguments (MIC, PK/PD index).

We thank the authors for this comment. As we have explained in the methodology section, PK/PD analysis was carried out by using Monte Carlo simulation.

As previously reported (Mouton JW, Clin Microbiol Infect. 2012; this reference appears in the reference list of the manuscript), real data defining the variability among individual patients are rarely available, and therefore, for PK/PD analysis, statistical approach is taken to simulate the variation. The statistical method most often used is Monte Carlo simulations. This method is a standard approach in the process of setting breakpoints and has been used by EUCAST since 2002. The simulated population present the same mean value and standard deviation than the original values of the pharmacokinetic parameters; therefore, results are reproducible.

  • The literature could be improved by the addition of relevant references.

The literature list has been improved, including the references suggested by the reviewer:

  • Huang C, Lin L, Kuo S. Risk factors for mortality in Stenotrophomonas maltophilia bacteremia - a meta-analysis. Infect Dis (Lond). 2024 May;56(5):335-347. doi: 10.1080/23744235.2024.2324365.
  • Gibb J, Wong DW. Antimicrobial Treatment Strategies for Stenotrophomonas maltophilia: A Focus on Novel Therapies. Antibiotics (Basel). 2021 Oct 9;10(10):1226. doi: 10.3390/antibiotics10101226.
  • Prawang A, Chanjamlong N, Rungwara W, Santimaleeworagun W, Paiboonvong T, Manapattanasatein T, Pitirattanaworranat P, Kitseree P, Kanchanasurakit S. Combination Therapy versus Monotherapy in the Treatment of Stenotrophomonas maltophilia Infections: A Systematic Review and Meta-Analysis. Antibiotics (Basel). 2022 Dec 9;11(12):1788. doi: 10.3390/antibiotics11121788.

Reviewer 2 Report

Comments and Suggestions for Authors

Authors have utilized a PK/PD approach to optimize antibiotics therapy for Stenotrophomonas malto- philia infections in critically ill patients. Although this article may be of interest to the readers, there are few things which are unclear. 

1. Most of the figures lack minimal resolution which makes it hard to understand them. 

2. Authors have stated that they have utilized PK/PD approach, however it is not clear what model schematic or type of model they have utilized. It is not clear whether they have performed compartmental modeling, Non compartmental analysis, mechanistic modeling. 

3. It is not clear which PD endpoint authors have utilized for optimization of therapy. 

Author Response

Point-by-point responses to reviewer 2 (changes in the manuscript are highlighted in red)

1. Most of the figures lack minimal resolution which makes it hard to understand them. 

In order to improve figure resolution, we have increased the size of the figures.

2. Authors have stated that they have utilized PK/PD approach, however it is not clear what model schematic or type of model they have utilized. It is not clear whether they have performed compartmental modeling, Non compartmental analysis, mechanistic modeling. 

Thanks to the reviewer for this comment. We have added this information in material and method section (page 16):

For trimethoprim, levofloxacin, minocycline, and tigecycline, fAUC24/MIC is the PK/PD index that best predict efficacy, and it was estimated by non-compartmental analysis using the following equation [20]:

For cefiderocol and aztreonam, the proportion of time that the unbound serum concentration remains above the MIC at steady-state (%fT>MIC) is the PK/PD index related to efficacy, and it was calculated considering one-compartmental pharmacokinetic model with the following equations [20]:

3. It is not clear which PD endpoint authors have utilized for optimization of therapy.

PD endpoints are the PK/PD targets, which are summarized in the table 3.

To better understand this, we have modified the following paragraph in the material and method section (page14):

The PK parameters of trimethoprim, levofloxacin, minocycline, tigecycline, cefiderocol, aztreonam, and avibactam in critically ill patients, as well as the PK/PD targets (PD endpoints), were obtained from literature. The PK parameters and the PK/PD targets are presented in Table 3.

Reviewer 3 Report

Comments and Suggestions for Authors

Your manuscript titled "Optimizing antibiotic therapy for Stenotrophomonas maltophilia infections in critically ill patients: a pharmacokinetic/pharmacodynamic approach" is well written and addressed an important clinical challenge: treating a multidrug resistant bacteria infection in critically ill patient.

The methods applied to optimize dosing plans are of great value.

The figures in the manuscript are of low resolution and are difficult to read.

Author Response

Point-by-point responses to comments of reviewer 3

The figures in the manuscript are of low resolution and are difficult to read.

We thanks the reviewer for the comments. In order to improve figure resolution, we have increased the size of the figures.

Round 2

Reviewer 2 Report

Comments and Suggestions for Authors

Authors have addressed my previous comments.